# The Influence of Microbial Fertilizers on the Rhizospheric and Epiphytic Microbiota, as Well as the Foliar Feeding Impact on Apple Leaf Mineral Contents

**DOI:** 10.3390/plants14243783

**Published:** 2025-12-12

**Authors:** Andrei I. Kuzin, Marina V. Maslova, Ludmila V. Stepantsova, Ivan N. Shamshin, Ekaterina V. Grosheva, Svetlana A. Karpukhina, Anastasiya A. Shmakova, Vladimir N. Nazarov, Vyacheslav N. Krasin, Natalia Ya. Kashirskaya, Anna M. Kochkina

**Affiliations:** 1I.V. Michurin Institute of Fundamental and Applied Agrobiotechnology, Michurinsk State Agrarian University, 393760 Michurinsk, Russia; marinamaslova2009@mail.ru (M.V.M.); stepanzowa@mail.ru (L.V.S.); ivan_shamshin@mail.ru (I.N.S.); ekaterina2687@mail.ru (E.V.G.); setka6808@mail.ru (S.A.K.); nastia.smakova.1996@yandex.ru (A.A.S.); vladimi.nazaroff@ya.ru (V.N.N.); krasin84@yandex.ru (V.N.K.); 2I.V. Michurin Federal Scientific Center, 393774 Michurinsk, Russia; kashirsckaya.n@yandex.ru (N.Y.K.); ms.anna.step@mail.ru (A.M.K.)

**Keywords:** apple tree, bacteria, endophytic and rhizosphere microbiota, foliar nutrition, fungi, leaf nutrient contents, microbial fertilizer

## Abstract

Analysis of the apple tree rhizosphere and phylloplane microbiota revealed the presence of pathogenic and conditionally pathogenic micromycetes: *Penicillium*, *Cladosporium*, *Fusarium*, *Mucor*, *Trichotecium*, *Alternaria*. The application of microbial fertilizers (MFs)—Azafok, Enzymocid, and Nitragin—reduced their abundance in the soil. This occurred due to the beneficial bacteria contained in the biopreparations (*Bacillus* spp., *Bacillus aryabhattai*, *Pseudomonas fluorescens*, *Bradyrhizobium japonicum*), which possess fungicidal activity and the ability to improve the mineral nutrition of plants, thereby enhancing their immune status. Nitragin also reduced the colonization of leaves by pathogenic fungi. The greatest reduction in contamination was achieved by the combined application of MFs with foliar feeding using mineral substances, particularly when using Azafok. The influence of MFs on the state of the epiphytic microbiota is associated with their indirect action through the activation of the host plant’s functional activity and the stimulation of its defense mechanisms. The MFs introduced into the soil stimulated an increase in the content of nitrogen, phosphorus, potassium, and calcium in the leaves. We also noted the influence of MFs on modifying the effect of foliar feeding on the nutrient content in the leaves. The leaf nitrogen content with the combined application of soil-applied Azafok and FF was lower than with soil application alone. Furthermore, the use of foliar fertilizing reduced the phosphorus and potassium content in the leaves against the background of Azafok and Enzymocid, although the relative level of these nutrient contents remained very high. Only the application of foliar fertilizing against the background of Nitragin stimulated an increase in the phosphorus and potassium content in the leaves. Further research is needed to clarify the nature of this modification.

## 1. Introduction

Ensuring food security for the growing global population is one of the most pressing challenges of the 21^st^ century. The world’s population is expected to exceed 9.7 billion people by 2050 [1], necessitating significant transformations in agriculture. It must change seriously to meet growing demand without causing environmental degradation. One of crucial components of this challenge should be the sustainable use of fertilizers without causing significant harm to the environment.

Consequently, the horticulture faces the dual challenge of enhancing orchard productivity while simultaneously reducing its negative environmental impact and improving orchard sustainability. This latter objective necessitates increased biodiversity and the broader adoption of nature-inspired technologies. A potential solution lies in reducing mineral fertilizer application rates by increasing its efficiency through soil enrichment with beneficial microorganisms, including both fungi and bacteria [2,3,4].

Microbial fertilizers are products containing beneficial live microorganisms, such as bacteria and fungi. By acting upon various substances present in the soil, these microbes convert nutrient elements into forms available to plants, thereby enhancing soil fertility [5]. A key distinction of these biofertilizers is that they improve soil fertility not only by increasing nutrient availability but also through the production of various growth-stimulating hormones and essential enzymes. They can also suppress pathogenic microorganisms in the soil.

Their application also affects the plant’s endophytic microflora, increasing its viability [6]. Unlike traditional chemical fertilizers, which provide nutrients only directly, microbial fertilizers create a favorable ecosystem within the soil, enhancing its fertility naturally and sustainably over the long term. Furthermore, the efficiency of mineral fertilizer application can be improved, allowing for a significant reduction in their application rates [7].

Results from numerous experiments have shown that biofertilizers enriched with mycorrhizal and thread-like fungi have a stimulatory effect on plant growth and reproduction [4]. Beneficial microorganisms assist roots in absorbing minerals from the soil and enhance the physiological activity of plants [[8][9]]. Microorganisms also play a vital role in maintaining long-term soil fertility and sustainability by fixing atmospheric nitrogen and converting unavailable soil macro- and microelements into forms accessible to plants [8].

At the same time, a number of constraints significantly reduce the effectiveness of microbial fertilizers. These include very low nutrient content in soils, high salinity, moisture deficiency, elevated temperatures, the presence of pests and diseases, competition with native microbial species, etc. [10,11], all of which adversely affect the viability and activity of the applied beneficial microorganisms. Since the application of such preparations affects plants not only through nutrient optimization but also in a more multifaceted way, the scope of research questions for studying their interaction with plant organisms can be significantly broader than in traditional studies with mineral fertilizers.

The literature contains substantial data on the influence of soil-applied fertilizers on soil microbiota [12,13,14]. There are also numerous publications on the impact of microorganisms (bacteria and fungi) on plant status [15]. For instance, bacteria of the genus *Bacillus* can stimulate plants to synthesize antimicrobial peptides, antioxidants, and other compounds that enhance protection against pathogens, as demonstrated in the case of bacterial wilt (caused by *Ralstonia solanacearum*) in tobacco plants [16]. Pseudomonas bacteria help plants develop induced systemic resistance through the synthesis of compounds such as salicylic acid and other volatile organic compounds [17,18]. Thus, microbial preparations can exert a multifaceted influence on plants, which is not necessarily directly related to mineral nutrition.

Foliar fertilizing triticale with nitrogen showed a stimulating effect on the total number of heterotrophic true bacteria, filamentous fungi, and ammonifying microorganisms, as well as the level of activity of acid and alkaline phosphatase, dehydrogenase, and urease. The presence of sulfur in the foliar fertilizer, increased the proliferation of soil microorganism cells after plant harvesting by over 100% and intensified the activity of soil enzymes, depending on their type, in the range of 15 to 100% [19]. There is also information that foliar application of salicilic acid (SA) influenced different bacterial genera in the rhizosphere, but the responsive genera varied between generations. No bacterial genera were detected that responded to SA application in the first generation suggesting that there are no immediate responses of bacteria in the rhizosphere to SA application to plants [20]. Thus, the influence of foliar fertilization can lead to specific genetic signals that affect the soil microbiome. In this case, it can be assumed that a reciprocal response may also occur—signals from the altered soil microbiota modify the effect of foliar fertilization on the plant.

One of the understudied areas, with insufficient information in the scientific literature, is the modulation of foliar fertilization impact on leaf nutrient status by soil-applied microbial fertilizers. We hypothesized that microbial fertilizers applied to the soil alter the overall physiological status of the plants by modifying the rhizosphere microbiota and the complex of influencing factors (e.g., the release of phytohormones, amino acids, peptides, etc.). This shift, in turn, modifies leaf activity and their receptiveness to foliar-applied nutrients.

The objectives of our study are: (i) to evaluate the effect of foliar fertilization on the content of macro- and microelements against the background of soil-applied microbial fertilizers; (ii) to assess the impact of soil-applied microbial fertilizers on the rhizosphere microbiota; (iii) to determine the influence of soil-applied microbial fertilizers on the leaf epiphytic microbiota. In our study, we investigated the effect of a complete foliar fertilization program against a background of soil applied beneficial microorganism products: the well-known Nitragin, based on beneficial organisms such as *Bradyrhizobium japonicum*, as well as the new microbial fertilizers Azafok, containing *Bacillus aryabhattai*, *Paenibacillus polymyxa*, *P. mucilaginosus*, and Enzymocid, which includes *B. subtilis*, *Pseudomonas fluorescens*, *Trichoderma harzianum*, and *Azotobacter chroococcum*.

## 2. Results

*Leaf nutrient contents*. In a number of treatments, we observed rather unexpected results from the application of microbial fertilizers, especially in combination with foliar feeding. For example, foliar feeding reduced the nitrogen content in the leaves when the fertilizer Azafok was applied, whereas in the treatment without foliar feeding, the leaf nitrogen content was significantly higher than in the unfertilized control 1 (Table 1). Conversely, with the application of the fertilizer Enzymocid, foliar feeding stimulated a substantial increase in leaf nitrogen content, which in this treatment was the highest in the experiment. Nitragin also provided high nitrogen content in the leaves, and in this case, foliar feeding enhanced the effect of the MFs application. It is important to note that the application of Nitragin caused the maximum increase in leaf nitrogen content in our experiment. At the same time, in the plots with the Nitragin treatments in early spring, we recorded the lowest concentations of soil hydrolyzable nitrogen (the initial contents of the soil primary nutrients are described in more detail in Section 4). This indicates the high efficiency of this fertilizer when spraying the soil with growing leguminous plants (white clover, mouse peas, etc.).

**Table 1 plants-14-03783-t001:** The influence of soil applied microbial fertilizers and foliar fertilizing (FF) on the contents of leaf primary nutrients, % d.m. (dry matter).

	Azafok		Enzymocid		Nitragin		Control		LCD_05_
	Soil	Soil + FF	Soil	Soil + FF	Soil	Soil + FF	Control 1	Control 2	
N	2.30 ^b^	1.96 ^a^	2.32 ^b^	3.19 ^e^	2.63 ^c^	2.88 ^cd^	2.02 ^a^	2.41 ^b^	0.27
P_2_O_5_	0.79 ^c^	0.62 ^b^	1.07 ^e^	0.84 ^cd^	0.58 ^b^	0.76 ^c^	0.47 ^a^	0.52 ^a^	0.09
K_2_O	3.30 ^b^	2.67 ^a^	3.81 ^c^	3.08 ^b^	2.54 ^a^	3.26 ^b^	2.62 ^a^	3.07 ^b^	0.39
Ca	0.82 ^cd^	0.84 ^cd^	0.77 ^bc^	0.82 ^cd^	0.87 ^cd^	0.82 ^cd^	0.64 ^a^	0.78 ^bc^	0.06
Mg	0.34 ^b^	0.50 ^e^	0.31 ^b^	0.42 ^cd^	0.20 ^a^	0.38 ^b^	0.30 ^b^	0.37 ^b^	0.04

Notes: different lowercase letters indicate significant differences among different treatments; The least significant difference (LSD) was calculated for comparisons between the different treatments at *p* < 0.05.

The lowest phosphorus content was found In the control treatments. In other words, the application of microbial fertilizers led to an increase in leaf nutrient content. However, in the control treatments, foliar fertilizing did not have a significant effect on the leaf nutrient content. In contrast, with the application of Enzymocid and Azafok, the leaf phosphorus content decreased significantly when foliar feeding was applied. Meanwhile, against the background of Nitragin application, the leaf phosphorus concentration increased significantly under the influence of foliar fertilizing.

The leaf potassium status changed In a roughly similar manner under the Influence of foliar feeding: it decreased with the soil application of Azafok and Enzymocid, but increased against the background of Nitragin use. In the treatment without soil fertilizer application (control 2), foliar fertilization led to a significant increase in potassium content.

In almost all treatments, the leaf calcium increased under the influence of foliar fertilizing (the exception was the Enzymocid treatment, where leaf fertilizing did not have a significant effect on the nutrient concentration).

The leaf magnesium content significantly increased due to foliar feeding in all treatments where soil fertilizers were applied. However, the application of microbial fertilizers without the use of foliar feeding had no visible effect on the leaf nutrient content compared to control 1.

Soil analysis revealed significant differences in nutrient content prior to the start of the experiment. To account for the influence of this variation on the leaf primary nutrient concentrations, the sample covariance for the soil nitrogen covariate (where the greatest differences were observed in spring) was −7.46 (population covariance −7.15; Pearson correlation r = −0.24). This indicates no significant relationship between the initial soil nitrogen contents and its leaf concentrations, a result could be attributed to the application of soil and foliar fertilizers. The sample covariance for soil phosphorus was 4.53 (population covariance 3.97; Pearson correlation r = 0.43), suggesting a moderate-strength relationship between the initial phosphorus contents and its leaf concentrations. This implies that the influence of applied fertilizers on soil phosphorus was less pronounced compared to their effect on nitrogen concentration. For the soil potassium covariate, the sample covariance was −3.02 (population covariance −2.64; Pearson correlation r = −0.64). This points to a complete absence of any relationship between initial soil potassium and its leaf concentration, which speaks to the substantial impact of the applied soil fertilizers and foliar feedings.

The leaf manganese Increased across all experimental treatments with the application of microbial fertilizers; however, a significant enhance was observed only with the application of Azafok and Enzymocid (Table 2). Foliar feeding increased the manganese concentrations in the leaves across all treatments with soil application of MFs.

Similarly, the zinc contents increased under the influence of foliar feeding on all studied soil backgrounds, although this increase was not always statistically significant. A substantial increase in zinc contents occurred only with the application of Enzymocid and Nitragin. We observed the maximum zinc content in the leaves with the soil application of Azafok, both with and without the use of foliar feeding. It should also be noted that with soil application alone without foliar fertilization, only the use of Azafok stimulated a significant increase in leaf zinc concentration compared to control 1.

**Table 2 plants-14-03783-t002:** The influence of soil applied microbial fertilizers and foliar fertilizing (FF) on the leaf micronutrient contents, mg kg^−1^ d.m.

	Azafok		Enzymocid		Nitragin		Control		LCD_05_
	Soil	Soil + FF	Soil	Soil + FF	Soil	Soil + FF	Control 1	Control 2	
Mn	40.92 ^b^	45.87 ^c^	42.93 ^c^	44.98 ^c^	38.77 ^ab^	43.48 ^bc^	36.92 ^a^	40.58 ^b^	3.37
Zn	18.74 ^cd^	19.98 ^d^	12.51 ^a^	15.61 ^b^	11.54 ^a^	17.62 ^c^	14.65 ^b^	15.81 ^b^	1.27
Cu	5.93 ^c^	6.04 ^c^	6.35 ^c^	6.05 ^c^	5.54 ^bc^	5.40 ^bc^	4.32 ^a^	4.96 ^b^	0.59
Mo	0.091 ^a^	0.135 ^b^	0.252 ^d^	0.153 ^c^	0.139 ^bc^	0.132 ^b^	0.134 ^b^	0.149 ^c^	0.014

Notes: different lowercase letters indicate significant differences among different treatments; The least significant difference (LSD) was calculated for comparisons between the different treatments at *p* < 0.05.

In the absence of soil fertilizer application, the copper content increased significantly in control 2 under the impact of foliar fertilizing. The application of microbial fertilizers also led to a significant increase in leaf copper concentration compared to control 1. However, foliar feeding, when used in treatments with soil-applied microbial fertilizers, did not have a noticeable effect on the leaf copper status.

The molybdenum content in plant leaves increased significantly only when Enzymocid was applied to the soil compared to the control 1. Future research should pay particular attention to the fact that foliar feeding reduced the molybdenum contents when Enzymocid and Nitragin were applied into the soil (this decrease was statistically significant only in the Enzymocid treatment). In contrast, by Azafok application the leaf molybdenum status increased significantly.

Plating the test suspension on agar reveals the dominant and competitive microorganisms in the soil and apple tree phylloplane. Numerous fungi and bacteria present in the experimental sample cannot grow on artificial media due to their physiological characteristics. So, the primary metric for comparing soil and epiphytic microbiota was the quantity of microorganisms that could develop colonies on the nutrient medium’s surface.

*Rhizosphere microbiota*. In all experimental treatments, the analyzed soil samples were colonized by both bacterial and fungal microbiota. The total microbial count was 181 × 10^4^ CFU/mL. Fungi accounted for 24.1% of all microorganisms, while bacteria assumed 75.89%. The micromycetes were represented by the following genera: *Penicillium* (40.96%), *Cladosporium* (22.98%), *Fusarium* (10.87%), *Mucor* (10.76%), *Trichoderma* (7.74%), *Trichotecium* (5.24%), and *Alternaria* (1.44%). Among these, only *Trichoderma* typically exerts a positive effect on plants. The remaining genera are either pathogenic or conditionally pathogenic fungi. The bacterial microbiota included both native strains and those introduced as part of the microbial fertilizers (MFs).

The composition of soil microorganisms In the apple orchard changed under the influence of the MFs (microbial fertilizers) applied to the root zone. An additional foliar treatment of the plants did not affect this parameter. In both treatment types (with and without foliar fertilizing) compared to the controls, an increase in bacterial colonization of the samples by 64.23% and 56.47%, respectively, was observed, which reduced the population of fungal microbiota by 19.56% and 15.51%, respectively.

We did not observe significant differences in the bacterial colonization of soil samples depending on which MFs was applied. In the treatments where only soil fertilizers were used, Enzymocid more actively reduced the number of conditionally pathogenic and pathogenic fungi than Azafok and Nitragin. The application of these preparations suppressed the growth of fungal pathogens by 37.83%, 30.25%, and 20.51%, respectively, compared to the control. No consistent patterns regarding the influence of foliar nutrition on the soil microbiota were identified (Figure 1).

*Epiphytic microbiota*. Analysis of the epiphytic microbiota of apple leaves across the different experimental treatments revealed the presence of mycelial fungi, yeasts, and bacteria in a percentage ratio of 67:8:25. The total colonization of the leaf surface averaged 186 × 10^4^ CFU/cm^2^ across all samples. Among the mycelial fungi, representatives of the genus *Cladosporium* dominated, accounting for 89.3% of all isolated micromycetes. *Penicillium* (6.40%), *Fusarium* (2.55%), and *Alternaria* (1.73%) were also encountered.

The soil application of MFs did not significantly affect the composition of the epiphytic microbiota on apple leaves. The abundance of the bacterial microbiota remained at a low level, ranging from 27.33 × 10^4^ to 44.00 × 10^4^ CFU/cm^2^, while in the control 1, this value was 40.33 × 10^4^ CFU/cm^2^. In the treatments with Azafok and Enzymocid without foliar feeding, no differences from the control 1 were observed in the fungal colonization of the leaf surface either. It is important to note that the most significant effect was observed with the combined application of MFs and foliar fertilization. The soil application of Azafok and Enzymocid in combination with foliar fertilizers led to a synergistic effect, promoting a significant activation of the epiphytic bacterial microbiota by 85.96% and 41.33%, respectively, compared to the untreated control. A reduction in the colonization by pathogenic fungi by 39.46% and 14.88%, respectively, was also noted.

Nitragin, while maintaining the bacterial population at the controls level, contributed to a 7.4% reduction in the fungal CFU on the leaf surface. The control 2 treatment with foliar fertilization (without MFs) and Nitragin + FF did not differ significantly from the untreated control 1 in terms of bacterial activity, but a reduction in fungal colonization by 7.00% and 16.41%, respectively, was noted. This fact may indicate that the FF, although it did not cause an active increase in the number of bacteria, promoted the activation of their functional activity and, as a result, enhanced the anti-fungal effect. Another reason for the reduction in the number of micromycetes could be the indirect influence of minerals on the phylloplane microbiota through the host plant tissues. The foliar fertilizer application improved plant nutrition and, consequently, its functional state, and thus its ability to combat pathogens (Figure 2).

## 3. Discussion

Our publication presents a report on the work completed over one year. The problem of reproducibility in research results is highly significant in science [21]. This study will be continued over the next two years, during which we plan to replicate the experiments conducted this season to draw accurate conclusions about specific mechanisms or metabolites that should be studied in future research, including by other researchers. At the same time, the results of our work appear very interesting, and we consider it our duty to present them to the scientific community.

The high concentrations of nitrogen in the leaves were obtained in the plots with the lowest initial nitrogen availability, where Nitragin (*Bradyrhizobium japonicum*) was applied, in both treatments (with and without foliar feeding). A comparable effect from inoculating with these microorganisms has been reported in the literature [22]. But something else is interesting: in the Azafok treatment with foliar feeding, the leaf nitrogen concentration decreased, while the application of Enzymocid stimulated the maximum leaf nitrogen status. Furthermore, foliar fertilizing increased the leaf nitrogen content in the control 1 plot without soil fertilizer application. In this case, this undoubtedly occurred due to the optimization of microelement contents, primarily manganese, as reported in the literature [23]. Thus, the effect of foliar applications of manganese-containing preparations was modified with the addition of Azafoc.

The foliar fertilization program we developed was primarily intended to optimize the calcium and microelement nutrition of apple trees. However, the results revealed a more complex pattern. Plants absorb micronutrients from foliar applications at different rates: the fastest is zinc uptake, then manganese, and boron [24]. An increase in leaf calcium content in response to foliar feeding was observed only in the control group (without soil-applied MFs). Among the microelements, the concentrations of only manganese and zinc showed a consistent increase across all soil treatment with foliar application (though these increases were not always statistically significant). In contrast, foliar feeding had no substantial impact on copper content in treatments where microbial biofertilizers were applied to the soil, even though a positive effect was clearly evident in the control 2. Similarly, the effect of foliar feeds on molybdenum content was inconsistent in the presence of different MFs, despite a significant increase in this nutrient being recorded in the control group with foliar application.

Enzymocid reduced pathogenic and conditionally pathogenic fungi in the soil by 37.83% (soil only), surpassing Azafok (30.25%) and Nitragin (20.51%) and this could be related to the modifying effect of microbial fertilizers on the impact of foliar fertilization. Zhang and colleagues [25] reported that root exudates change according to the plant’s growth and development stages, and at the same time they can actively influence the composition of the soil microbiome, specifically the proliferation of *B. subtilis* [26]. The literature contains reports on how root exudation allows plants to regulate nitrogen uptake in soils with varying availability of this nutrient [27]. Additionally, through the release of exudates and symbiosis with arbuscular mycorrhizal fungi, plants can regulate phosphorus uptake [28]. Root exudates also play an important role in the uptake of other macroelements (nitrogen, phosphorus, potassium, calcium, magnesium, and iron [29], and changes in root exudates undoubtedly influence the concentration of nutrients in the leaves. Due to the interaction of fertilizers applied to the soil with the plant and their effect on the leaf microflora, changes in the absorption of nutrients from foliar fertilizers are likely, ащк example the application of Azafok in our study demonstrated the greatest synergistic effect in activating epiphytic bacteria (85.96%) and reducing foliar fungi (39.46%).

Based on our research findings, it can be concluded that the expected effect of foliar fertilizers on the leaf nutrient contents was not observed in all treatment. The alteration in foliar fertilizer’s effect on nutrient levels is connected to changes in the rhizosphere’s microbiota and metabolites due to soil application of MFs. The acceleration of growth processes in apple trees under the influence of MBs, resulting from the production of growth-stimulating substances by the bacteria they contain, could have led to a decrease in the concentration of some nutrients due to a dilution effect. The features of microorganism-environment interactions are insufficiently studied. In particular, there are aspects that cannot be elucidated by plant scientists and microbiologists alone but can be investigated with the help of representatives from other scientific disciplines [30].

In a laboratory conditions, soil microbiome analysis showed bacteria were more dominant than fungi. Furthermore, the majority of the isolated micromycete colonies were classified as pathogenic and conditionally pathogenic fungi, as well as toxin producers. These included *Penicillium*, *Cladosporium*, *Fusarium*, *Mucor*, *Trichotecium*, and *Alternaria*. The presence of these microorganisms in the soil of an apple orchard is undesirable, as it can lead to the accumulation of toxins in the substrate, as well as to associative root infection, followed by damage to the plant’s vascular system. This is particularly likely when the host is weakened due to the adverse effects of abiotic stress factors. Studies by many researchers note that the presence of this group of fungi is one of the primary reasons leading to the occurrence of Apple Replant Disease (ARD) [31,32,33,34].

In order to prevent the accumulation of these micromycetes in the soil, it is advisable to ensure that the physiological condition of plants is at an optimal level and to implement agricultural practices that strengthen their immune systems while mitigating biotic stressors. A promising approach in this regard is the use of biopreparations containing alive bacterial structures that improve the growth and mineral nutrition of plants and possess fungicidal and fungistatic properties. These species include *Bacillus aryabhattai* [35]; *Bacillus subtilis* [36]; *Paenibacillus polymyxa* [37]; *Paenibacillus mucilaginosus* [38]; *Pseudomonas fluorescens* [39]; *Bradyrhizobium japonicum* [40].

As a result, a rise in the non-pathogenic bacteria microbiota in the soil of an apple orchard is a positive indicator linked to its fertility. To achieve this, the soil microbial community is enriched with symbiotic bacteria. For this purpose, the microbial fertilizers Azafok, Enzymocid, and Nitragin were used.

Informative indicators of orchard soil health are the microorganisms that inhabit it. Studying the quantitative and qualitative composition of the rhizosphere microbiota makes it possible to assess the effectiveness of the agrotechnical measures applied for managing soil fertility [41].

The conducted studies indicate that the use of MBs containing a complex of bacteria with nitrogen-fixing, phosphate- and potassium-solubilizing, as well as fungicidal properties in the apple orchard, generally led to a reduction in the population of fungal pathogens and an accumulation of bacteria. Among the tested MFs Enzymocid was the most effective, compared Azafok and nitrogin, suppressing fungi and boosting bacteria.

Among the epiphytes of apple leaves, mycelial fungi dominated over bacteria and yeasts. The most competitive were representatives of the genera *Cladosporium*, *Penicillium*, *Alternaria*, and *Fusarium*, which have either a pathogenic or conditionally pathogenic impact on the plant. The similarity in the qualitative composition of the epiphytic and rhizosphere microbiota is explained by the fact that phyllosphere microorganisms primarily travel there via air, and they are carried into the air from the soil [42].

The conducted studies have shown that the most pronounced effect of bacteria from MFs is observed during their direct interaction with the microorganisms of the rhizosphere. In this case, the biocontrol agents that are part of the microbial fertilizers directly affect the plant microbiota through the synthesis of metabolites with anti-fungal action, shifting the balance between micromycetes and bacteria.

The action of MFs can also be carried out indirectly through the host plant by altering its physiological state and immune status. As a consequence, the composition of root and leaf exudates changes, their rate of secretion also alter due to the modulation of osmotic processes and stomatal function. This, in turn, influences the microbiota, which in turn affects the microbiota [42,43,44]. The application of MFs impacts all microorganisms interacting with the host, including indirectly related ones.

It has been proven that fertilization can significantly alter the composition and diversity of a plant’s microbial community [45,46,47]. An increase in nutrient availability correlates with an acceleration of microorganism growth and an improvement in substrate metabolism [48]. These assertions were formulated primarily to characterize processes occurring in the soil. However, the same can be said for the epiphytic microflora of leaves, as our research has shown.

Foliar fertilizing and epiphytic microflora interact in a complex manner. The foliar application of fertilizers influences the plant’s nutritional status and leaf surface properties, which, in turn, alters the composition and function of the epiphytic microbial community. Foliar fertilizers are a direct source of nutrients that can be absorbed by the leaves, affecting both the plant and the microbes, and may be most effective when combined with soil fertilizers to improve the plant’s functional state. Mineral elements directly influence the plant’s immune status by modulating the activity of redox enzymes, the secretion of plant exudates, the biosynthesis of phytoalexins, and the population dynamics of the microbiota [49].

Among the studied biopreparations, Nitragin positively influenced the host plant’s defense functions against pathogens, which was reflected in a reduced contamination level of apple leaves. When using foliar fertilizers, the population density of micromycetes in the phylloplane decreased most significantly in the treatments with Azafok. At the same time, the effectiveness of Enzymocid and Nitragin alone was comparable.

The application of microbial fertilizers Azafok, Enzymocid, and Nitragin alters the composition of the rhizosphere microbiota by increasing the bacterial population and reducing the proportion of pathogenic fungi. Nitragin was also able to influence phylloplane microorganisms, reducing fungal contamination. The greatest effectiveness was demonstrated by the combined application of MFs and foliar fertilizers with conventional agrochemicals. In this regard, Azafok proved promising, as it, together with NP mineral substances, significantly stimulated the accumulation of epiphytic bacteria and also reduced contamination by pathogenic fungi.

The plants’ response to treatments with zinc-containing agrochemicals applied as foliar feeds is also noteworthy. Foliar feeding increased the leaf zinc contents across all treatments. However, when Azafok and Enzymocid were applied to the soil, the leaf potassium status decreased in the treatments with foliar fertilization. In contrast, in the absence of soil fertilizer (control 2) and with the application of nitragin, the leaf potassium contents increased under foliar sprayings. According to earlier research, foliar application of zinc increases potassium uptake and leaf status [22,50,51]. We can speak of a certain modification of the foliar fertilization impact in this case as wells. One of the reasons for this phenomenon could be the alteration of the leaf’s epiphytic microflora and the overall physiological status of the plant under the influence of the soil-applied microbial fertilizers. The influence of soil applied MFs manifests not only through the suppression of soil pathogens due to the fungicidal activity of beneficial microorganisms, but also through the modulation of various physiological processes in the host plant (transpiration, photosynthesis, immune responses, etc.), which leads to changes in its mineral status.

## 4. Materials and Methods

Study site and experiment design. The study was conducted in an apple orchard planted with the cultivar ‘Rozhdestvenskoe’ at the JSC “Dubovoe” (52.622551, 40.285713). The trees were grafted on B396 rootstock. The orchard was established in 2012 with a planting pattern 4.5 × 1.45 m (1533 trees ha^−1^), drip irrigation system was installed. The experiment was set up in a randomized complete block design with three replications. Each repetition contained 50 trees, resulting in 150 trees per treatment. The total area of the experimental site was 2 ha. Experimental treatments: 1. Control 1 (No soil fertilizer, No foliar feeding (FF)); 2. Control 2 (No soil fertilizer, +FF); 3. Azafok; 4. Azafok + FF; 5. Enzymocid; 6. Enzymocid + FF; 7. Nitragin; 8. Nitragin + FF. For the mineral nutrition study, 3 trees were randomly selected from each replicate plot (15 trees per treatment total). Leaves (100 per replication) and rhizosphere soil (0–20 cm deep) were subsequently collected from these trees. The following microbial fertilizers (MFs) were applied to the soil: Azafok: 4 L ha^−1^ (*Bacillus aryabhattai* BR4 (VKPM B-3579), *Paenibacillus polymyxa* (VKM B-747), *Paenibacillus mucilaginosus* 27 (VKPM B-13582); total titer 1 × 10^9^ CFU mL^−1^); Enzymocid: 0.2 kg ha^−1^ (*Bacillus subtilis*, *Pseudomonas fluorescens*; *Trichoderma harzianum*; *Azotobacter chroococcum*; total titer 3 × 10^6^ CFU g^−1^); Nitragin: 0.5 L ha^−1^ (*Bradyrhizobium japonicum*; total titer 1 × 10^9^ CFU mL^−1^).

The microbial fertilizers were applied three times during the growing season using an OP-P-G 2000 (Russia) orchard herbicide sprayer. Foliar feeding was carried out in tank mixtures with plant protection products using an SLV-2000 (Moldova) garden sprayer. The foliar fertilization program is detailed in Table 3.

**Table 3 plants-14-03783-t003:** Foliar fertilizing program.

Phenological Growth Stage [52]	Acrochemicals	Application Rate. L ha^−1^
10 (Mouse-ear stage: Green leaf tips 10 mm above the bud scales)	Biostim Growth Ultramag Super Zinc-700	1.5 1.0
53 (Bud burst: green leaf tips enclosing flowers visible)	Akh-576-23 Ultramag Boron	1.5 1.0
57–61 (Pink bud stage: flower petals elongating; sepals slightly open; petals just visible—Beginning of flowering: about 10% flowers open)	Akh-576-23 Ultramag Boron	1.0 1.5
69 (End of flowering: all petals fallen)	Biostim Growth Ultramag Super Sulfur-900	1.0 2.0
71 (Fruit size up to 10 mm; fruit fall after flowering)	Ultramag Calcium Active	3.0
72 (Fruit size up to 20 mm)	Akh-576-23 Ultramag Calcium Ultramag Potassium	1.5 1.5 1.5
73 (Second fruit fall)	Biostim Growth Ultramag Potassium	1.5 1.5
74 (Fruit diameter up to 40 mm)	Akh-577-23 Ultramag Calcium	2.0 3.0
75 (Fruit about half final size)	Akh-577-23 Ultramag Calcium	1.0 3.0
76 (Fruit about 60% final size)	Ultramag Calcium	2.0
77 (Fruit about 70% final size)	Ultramag Calcium	3.0
81 (Beginning of ripening: first appearance of cultivar-specific colour)	SC2020	3.0

The composition of foliar fertilizers is in Table 4.

**Table 4 plants-14-03783-t004:** The contents of nutrients in agrochemicals for foliar fertilizing, % a.i. (active ingredients).

	N	P_2_O_5_	K_2_O	CaO	MgO	SO_3_	B	Cu	Fe	Mn	Mo	Zn	Amino Acids of Plant Origin
Biostim Growth	4.0	10.0			2.0	1.0	0.1		0.4	0.2		0.2	4.0
SC2020				10.0									
Ultramag Boron							11.0						
Ultramag Calcium	10.0			17.0	0.8		0.05	0.02			0.001	0.02	
Ultramag Calcium Active				10.0									
Ultramag Potassium	2.6		22.0										
Ultramag Super Sulfur-900	5.0					70.0							
Ultramag Super Zinc-700	1.5											40.0	
AKh-576-23	4.6				4.0		0.5	0.05	1.0	4.0	0.05	3.0	
AKh-577-23	2.0			10.0	5.0				0.7	2.0	0.04		

*Soil and leaf analysis*. The soil was analyzed for the contents of hydrolysable nitrogen using the Kjeldahl method (AKV-20, JSC Villitek, Moskow, Russia); mobile phosphorus via blue coloration and photocalorimetry (Hitachi U2000 spectrophotometer, Hitachi Ltd., Tokyo, Japan); exchangeable potassium (FPA-2-01 flame photometer, JSC ZOMZ, Zagorsk, Russia); and pH in KCl extract (Expert-001 pH-meter, Econics-Expert, Moscow, Russia). The soil analysis was performed before the establishment of the field experiment. For plant nutrition assessment, leaves were sampled on 22 June (stage 74, fruit diameter up to 40 mm). The leaves were analyzed for the contents of total nitrogen (using the Kjeldahl method, AKV-20, JSC Villitek, Moskow, Russia); molybdenum (by atomic absorption spectrometry, MGA-915MD, Saint Petersburg, Lumex, Russia); and phosphorus, potassium, calcium, magnesium, manganese, zinc, and copper (via X-ray fluorescence spectrometry, ADC Prisma M, Yuzhpolymetal-Holding Group, Moscow, Russia) [53].

*Soil characteristics*. The contents of soil primary nutrients across the treatments can be characterized as medium to high (Table 5). The soil acidity can also be assessed as favorable for apple cultivation. The content of hydrolysable nitrogen varied the most across the experimental plots. In some treatments, the soil nitrogen was relatively low, while in others, it was high. The content of mobile phosphorus was less variable, and the levels of exchangeable potassium and soil acidity fluctuated even less.

**Table 5 plants-14-03783-t005:** The baseline values of soil nutrients N, P, K, and pH of the plots designated to receive these treatments (before the experiment, 31 March 2025).

	N Hydrolysable. mg kg ^−1^	P_2_O_5_ Mobile. mg kg ^−1^	K_2_O Exchangeable. mg kg ^−1^	pH_kCl_
Control 1 (without FF)	128.2	210.8	156.1	5.49
Control 2 (FF)	100.2	190.1	165.8	5.29
Azafok (without FF)	171.0	156.9	164.3	5.86
Azafok (FF)	193.4	201.8	164.5	5.82
Enzymocid (without FF)	181.0	259.5	133.3	5.86
Enzymocid (FF)	209.4	235.5	148.0	5.21
Nitragin (without FF)	108.6	259.5	159.6	5.55
Nitragin (FF)	114.2	210.8	152.4	5.39

*Isolation of rhizosphere microbiota*. For each experimental treatment, soil was collected as 3 composite samples, each consisting of 3 sub-samples taken at a depth of 17–20 cm from the apple tree root zone. For the analysis, a soil suspension was prepared by mixing 20 mL of soil with 180 mL of sterile distilled water in 250 mL Erlenmeyer flasks. The suspension was incubated on a shaker for 40 min at 190 rpm. Then, 10 mL of this suspension was taken and diluted in 90 mL of sterile distilled water. From the final suspension, 100 µL was plated in triplicate onto Petri dishes with potato dextrose agar and spread evenly over the surface using a spatula.

*Isolation of leaf epiphytic microbiota.* For each experimental treatment, three composite leaf samples were prepared. Each composite sample, in turn, consisted of three sub-samples. Leaf sample for forming a sub-sample was conducted randomly from a single tree, taking 10 leaves. From these, leaf discs 1.5 × 3.0 cm were cut and placed in flasks containing 100 mL of sterile distilled water. The flasks were incubated in a shaker for 40 min at 190 rpm. Each composite sample was prepared by combining three equal parts from the sub-samples. Subsequently, 100 µL of the suspension was plated in triplicate onto Petri dishes containing an agar-solidified potato-glucose medium, and spread evenly over the surface using a spatula. To suppress bacterial growth in the Petri dishes where fungal colonies were counted, Potato Dextrose Agar was supplemented with an antibiotic.

Following the plating of explants and the test suspensions, regular monitoring was conducted. The appearance of microbial colonies was noted, followed by identification via cultural, morphological features, and microscopy.

Subsequently, a recalculation of the Colony-Forming Units (CFU) was performed: for the rhizosphere microbiota, the count was expressed per 1 mL of soil, and for the epiphytic microbiota, it was expressed per 1 cm^2^ of leaf surface.

*Statistical analysis of the results.* The data were processed using Fisher’s analysis of variance method (ANOVA), we also conducted an analysis of covariance (ANCOVA) and calculated the Pearson correlation [54]. The least significant difference (LSD) was calculated for comparisons between the different treatments at *p* < 0.05. Differences that exceeded the calculated LSD value were considered statistically significant.

## 5. Conclusions

In conclusion, our findings indicate that the effect of foliar feeding on the nutrient content in leaves was not consistent across all treatment combinations and did not always align with initial expectations. We propose that this modification of the plant response on fertilizer application is intrinsically linked to alterations in the rhizosphere microbiota and its metabolic profile resulting from the soil application of microbial biofertilizers. Our results support a model of complex interaction between foliar applications and the epiphytic microflora. So, our study demonstrated the first evidence of differential modification in leaf N, P, and K content when FF is combined with specific FMs. Maximum effectiveness of fertlization across the set of traits was observed when FMs (such as Azafok) were combined with conventional foliar agrochemicals. The impact of soil-applied microbial fertilizers is manifested not only through the suppression of soil pathogens via the fungicidal activity of beneficial microorganisms but also through the modulation of the plant’s immune status, resulting from improved mineral nutrition The application of soil fertilizers alters the plant’s nutritional status and the physicochemical properties of the leaf surface (the phyllosphere). This creates a feedback loop, subsequently reshaping the composition and functional potential of the epiphytic microbial community. Foliar fertilizers provide a direct nutrient source for both plant uptake and microbial utilization on the leaf surface. And it amplifies the effects, contributing to a further enhancement of the plants’ functional state. But their effectiveness is highly context-dependent, influenced by the existing soil and plant microbiome. Taken together, this explains why the positive results from foliar fertilization are not always consistently observed and depend on the complex interaction of agronomic, microbial, and physiological factors. This suggests that foliar nutrition strategies are most effective not in isolation, but as an integrated component of a holistic fertilization program, combined with soil-applied amendments to optimize overall plant health and quality of fruits. However, further research is required to more precisely elucidate all aspects of this phenomenon.

## Figures and Tables

**Figure 1 plants-14-03783-f001:**
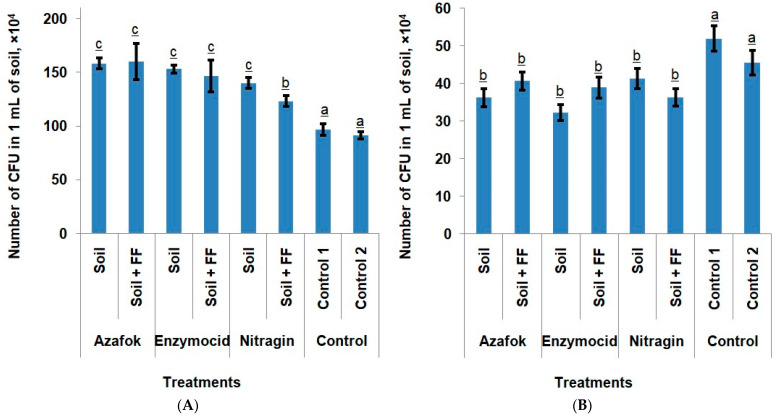
Bacterial (**A**) and fungal (**B**) colonization of soil samples in the apple orchard under soil application of microbial preparations and mineral foliar fertilizers (FF), (Notes: different lowercase letters indicate the treatments with statistically confirmed differences).

**Figure 2 plants-14-03783-f002:**
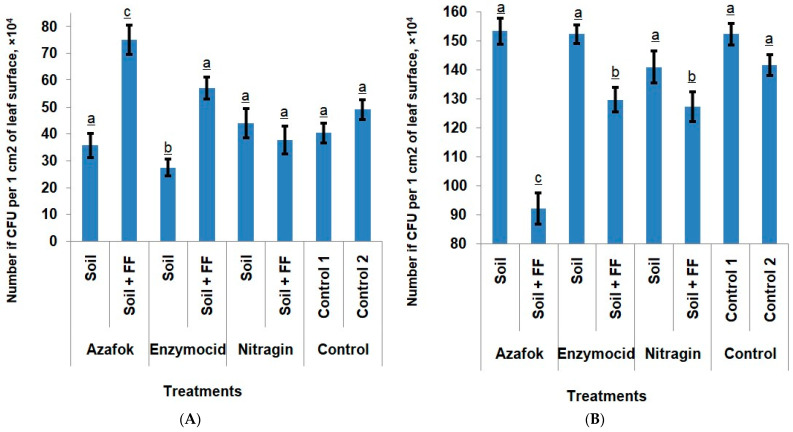
Bacterial (**A**) and fungal (**B**) colonization of apple leaves using microbial preparations and foliar fertilizers (FF), (Notes: different lowercase letters indicate the treatments with statistically confirmed differences).

## Data Availability

All the data generated during the study are presented within the article. The raw data are available from the corresponding author on reasonable request.

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
