# Peer review of "The Influence of Microbial Fertilizers on the Rhizospheric and Epiphytic Microbiota, as Well as the Foliar Feeding Impact on Apple Leaf Mineral Contents"

_plants, 2025, doi:10.3390/plants14243783_

Round 1

Reviewer 1 Report

Comments and Suggestions for Authors

The manuscript study the influence of microbial fertilizers on the rhizospheric and epiphytic microbiota, and the foliar feeding impact on apple leaf mineral contents. The study explore the interactions between soil-applied microbial fertilizers, foliar nutrition, and plant-associated microbiota. The topic is of interest, the experimental design is generally sound, the data presented is valuable, and the findings have clear practical applications for sustainable horticulture. Some points should be improved as following.

Title and text, "microbiological fertilizers" - Consider using "microbial fertilizers" consistently.

The introduction could be more focused. The link between soil MFs and the modulation of foliar nutrient uptake, which is the novel aspect of this paper, could be introduced earlier and more prominently.

I couldn't find a citation for reference 5 in the text.

Line 53, change "affects also" to "also affects".

Tables 1 and 2 are confusing and contain errors. Table 2 appears to have a duplicated column header block in the middle of the data rows, making it very difficult to read. The superscript letters (e.g., a, b, c) denoting statistical significance are present but not explained in the table captions. LCD05 also need to explain.

Line 120, delete “in leaves”.

Line 179, change “Fig. 1, 2” to “Figure 1, 2”.

The connection between the rhizosphere microbiota changes and the leaf nutrient/epiphyte changes is currently asserted but not strongly supported by the presented data. The discussion should more carefully acknowledge this and propose specific mechanisms or metabolites that could be investigated in future work.

The discussion is somewhat descriptive and repeats many findings from the results. It needs to be more synthetic and analytical.

Table 5 shows significant variation in initial soil nutrients (especially nitrogen) across plots before the experiment. This is a major confounding factor. The authors must explicitly address how this pre-existing variation was accounted for in the statistical analysis and the interpretation of the results, particularly for leaf nitrogen content. An analysis of covariance with initial soil values as a covariate would have been ideal; if not performed, this must be stated as a limitation.

Author Response

Dear Reviewer 1, we are deeply grateful to you for your work on our manuscript.

Reviewer 1

The manuscript study the influence of microbial fertilizers on the rhizospheric and epiphytic microbiota, and the foliar feeding impact on apple leaf mineral contents. The study explore the interactions between soil-applied microbial fertilizers, foliar nutrition, and plant-associated microbiota. The topic is of interest, the experimental design is generally sound, the data presented is valuable, and the findings have clear practical applications for sustainable horticulture. Some points should be improved as following.

RESPONSE:    dear Reviewe 1, we appreciate the careful analysis and positive evaluation of our manuscript by the reviewer. Please see below our detailed responses to the comments made/questions raised by the reviewer.

  1. Title and text, "microbiological fertilizers" - Consider using "microbial fertilizers" consistently.

RESPONSE: we agree with Reviewer 1 and have replaced the term "microbiological fertilizers" with "microbial fertilizers" throughout the text. (highlighted in yellow).

  1. The introduction could be more focused. The link between soil MFs and the modulation of foliar nutrient uptake, which is the novel aspect of this paper, could be introduced earlier and more prominently.

RESPONSE: we agree with this point. The text in the Introduction has been revised as suggested (see yellow highlights).

  1. I couldn't find a citation for reference 5 in the text.to molybdenum content, or cite relevant literature.

RESPONSE: thank you for catching this. The typo has been fixed. (highlighted in yellow).

  1. Line 53, change "affects also" to "also affects".

RESPONSE: thank you for pointing this out. The correction has been made (highlighted in yellow).

  1. Tables 1 and 2 are confusing and contain errors. Table 2 appears to have a duplicated column header block in the middle of the data rows, making it very difficult to read. The superscript letters (e.g., a, b, c) denoting statistical significance are present but not explained in the table captions. LCD05 also need to explain.

RESPONSE: this duplication was a technical error in the final editing phase and we corrected. We also have corrected the tables by adding explanatory footnotes for the superscript letters and for the LSD05 values. The LSD05 is also described in the Materials and Methods (highlighted in yellow).

  1. Line 120, delete “in leaves”.

RESPONSE: we corrrected this

  1. Line 179, change “Fig. 1, 2” to “Figure 1, 2”.

RESPONSE: thank you for the clarification. This has now been done. (see yellow highlights).

  1. The connection between the rhizosphere microbiota changes and the leaf nutrient/epiphyte changes is currently asserted but not strongly supported by the presented data. The discussion should more carefully acknowledge this and propose specific mechanisms or metabolites that could be investigated in future work.

RESPONSE: this publication is a communication reporting the findings of a one-year study. The issue of research reproducibility is of critical importance in science (White et al., 2025). The present study is part of an ongoing project scheduled to continue for the next two years. During this period, we plan to replicate the experiments conducted this year to strengthen our conclusions and identify specific mechanisms or metabolites that warrant more detailed investigation in future research, including by other teams. We nevertheless consider our preliminary results to be highly interesting and believe it is our responsibility to share them with the scientific community, even in the concise format of a communication, rather than withholding them until a full study will done, because for us is important the reaction of other scientists on our results.

  1. The discussion is somewhat descriptive and repeats many findings from the results. It needs to be more synthetic and analytical.

RESPONSE: we have addressed the comments of Reviewer 1 and revised the Discussion section accordingly (changes are highlighted in yellow).

  1. Table 5 shows significant variation in initial soil nutrients (especially nitrogen) across plots before the experiment. This is a major confounding factor. The authors must explicitly address how this pre-existing variation was accounted for in the statistical analysis and the interpretation of the results, particularly for leaf nitrogen content. An analysis of covariance with initial soil values as a covariate would have been ideal; if not performed, this must be stated as a limitation.

RESPONSE: we thank Reviewer 1 for this important suggestion. Following the recommendation, we performed covariance and correlation analyses based on the available data and have supplemented the Results section accordingly (changes are highlighted in yellow).

Reviewer 2 Report

Comments and Suggestions for Authors

Dear Authors,

Suggestions for correction are described in the attached file.

Kind regards,

Author Response

Dear Reviewer 2, we are deeply grateful to you for your work on our manuscript.

Reviewer 2

Dear Reviewer 2, we appreciate the careful analysis and positive evaluation of our manuscript by the reviewer. Please see below our detailed responses to the comments made/questions raised by the reviewer.

  1. Dear Authors, regarding the Abstract, I had the following question: did FF reduce the nitrogen content in certain treatments (e.g., with Azafok), or did it reduce P and K with Azafok and Enzymocid, while increasing these nutrients with Nitragin?.

RESPONSE: the leaf nitrogen content with the combined application of soil-applied Azafok and FF was lower than with soil application alone. Furthermore, the use of foliar fertilizing reduced the phosphorus and potassium content in the leaves against the background of Azafok and Enzymocid, although the relative level of these nutrient contents remained very high. Only the application of foliar fertilizing against the background of Nitragin stimulated an increase in the phosphorus and potassium content in the leaves.We added the corresponding information to the abstract. (highlighted in yellow).

  1. In the keywords you included the word "endophyric", please verify that the standard term accepted in the literature is "endofitico" to avoid ambiguity or the perception of a typo.

RESPONSE that was an unfortunate typo, and we have corrected it. (see yellow highlights).

  1. In the introduction, it would be important to mention that the pharmaceutical ingredients used (Azafok, Enzymocid, Nitragin) contain beneficial microorganisms.

RESPONSE: we have added the corresponding information to the introduction section (highlighted in yellow).

  1. Shouldn't the "Materials and Methods" section now be the correct one?

RESPONSE: dear Reviewer 2, according to the article template for the Plants journal, the Materials and Methods section is placed after the Discussion section.

  1. Dear authors, in the Results, you mentioned that Enzymocid reduced pathogenic and conditionally pathogenic fungi in the soil by 37.83% (soil only), surpassing Azafok (30.25%) and Nitragin (20.51%). This data should be highlighted in the discussion to justify the differences in performance of the fungicide.

RESPONSE: thank you for your recommendations. We have added the corresponding information to the Results sections (highlighted in yellow).

  1. In the captions for Tables 1 and 2, include a footnote explaining that different superscript letters (a, b, c, etc.) within the same line indicate statistically significant differences determined by p < 0.05.

RESPONSE: we have corrected the tables adding explanatory footnotes for the superscript letters and for the LSD05 values. The LSD05 is also described in the Materials and Methods (highlighted in yellow)

  1. Table 2 presents a redundancy in the column structure. The micronutrient symbols (Mn, Zn, Cu, Mo) and the data columns are repeated in the same table, which impairs readability and formatting. This repetition should be eliminated.

RESPONSE: we have eliminated this repetition. It was some sort of unfortunate glitch during the final copying.

  1. Ensure that the textual description of the CFU counts is fully aligned with the y-axis labels in Figures 1 and 2, to avoid any misunderstanding regarding the order of magnitude.

RESPONSE: we have eliminated possible misunderstandings in the text (highlighted in yellow).

  1. Dear authors, regarding the discussion of the data, although Enzymocid was the most effective in reducing soil pathogens (37.83%), Azafok demonstrated the greatest synergistic effect in activating epiphytic bacteria (85.96%) and reducing foliar fungi (39.46%). This contrast is fundamental to justifying the final conclusion that Azafok is the "most promising".

RESPONSE: thank you for the remark. We have clarified what specifically we consider promising regarding the application of Azafox (changes are highlighted in yellow).

  1. In the discussion you mention that the alteration in FF (foliar production) may be due to changes in the epiphytic microflora and the physiological status of the plant. Based on cited references regarding the action of Bacillus and Pseudomonas, how could the alteration in root and leaf exudates lead to contradictory observations of macronutrients (e.g., the reduction of K under Azafok+FF)?

RESPONSE: dear Rewiever 2, we have added the corresponding references to the sources in the literature in the discussion (changes are highlighted in yellow), but we cannot provide a definitive answer to the question "How exactly does this happen?" since the study of root exudates was not within the scope of our research.

  1. Dear authors, in the Materials and Methods section you mention that for the isolation of rhizosphere microbiota, you used potato dextrose agar (generally for fungi) and for epiphytic microbiota, you mention "agar-based nutrient medium". Since the study quantifies both bacteria and fungi in both compartments, it is essential to clearly specify which selective or non-selective media were used for counting bacteria and fungi in each sample (soil and leaf).

RESPONSE in the studies, a non-selective medium, Potato Dextrose Agar (PDA), was used. However, to suppress bacterial growth in the Petri dishes where fungal colonies were counted, Potato Dextrose Agar supplemented with an antibiotic was used. A comparative analysis of the experimental variants was conducted based on the number of CFU (Colony-Forming Units) of microorganisms cultured on the artificial nutrient medium, Potato Dextrose Agar. We have also added this information to the Materials and Methods section (highlighted in yellow).

  1. The leaves were sampled on June 22. It is recommended to indicate which phenological stage corresponds to that date, considering that FF occurred at various stages (Table 3), to contextualize the nutritional results. In the titles of the Tables and Figures, it is important to maintain the nomenclature Control 1 (Without FM, Without FF) and Control 2 (Without FM, + FF) to facilitate interpretation.

RESPONSE: we have add corresponding information in Materials and Methods section (highlighted in yellow).

  1. Table 5 lists the soil nutrient content before the experiment (March 31, 2025), but includes treatments with and without FF (e.g., "Azafok without FF" and "FF"). To avoid confusion, the legend should explicitly specify that these are the baseline values of N, P, K, and pH of the plots designated to receive these treatments, confirming the initial heterogeneity (especially of N).".

RESPONSE: thank you for the remark. We have changed the title of Table 5 according to the recommendation (highlighted in yellow).

  1. Dear authors, please include in the Conclusion that the study demonstrated the first evidence of differential modification in leaf N, P, and K content when FF is combined with specific FMs, which will allow you to differentiate your study from other traditional works.?

RESPONSE: we have added the corresponding information to the conclusion (highlighted in yellow).

  1. It is important that you report that maximum effectiveness was observed when FMs (such as Azafok) were combined with conventional foliar agrochemicals.

RESPONSE: we have added the corresponding information (highlighted in yellow).

  1. Dear authors, in the Conclusions you can also explain how the alteration in the rhizosphere led to contradictory foliar nutritional responses.

RESPONSE: dear Reviewer 2, we are not prepared to provide such an explanation in full, as we believe we must be cautious with such conclusions. We are confident that it involves, among other things, the transmission of certain signals within the plant organism at the genetic and hormonal levels; however, we are not conducting such studies.

Round 2

Reviewer 1 Report

Comments and Suggestions for Authors

The manuscript was revised according to the Reviewers’ comments. I think it can be accepted after minor revision. Some points should be improved as following.

The table should be revised as “three-line table”.

Table 2: the values of Control 1, Control 2, LCD05 (Column 8, 9, 10) are similar with the values of Soil, Soil +FF, Soil (Column 2, 3, 4). Please check the data.

Figure 1 and 2: please add the results of statistical analysis, e.g. mark different lowercase letters indicating significant differences among different treatments.

Author Response

Dear Reviewer, thank you once again for your work in improving our manuscript.

We have incorporated your suggested changes into the article:

we formatted the tables as "three-line tables,"

clarified the data in Table 2 and corrected columns 8, 9, and 10;

we also added to Figures 1 and 2, using different lowercase letters to indicate significant differences among treatments.
